# SapC–DOPS as a Novel Therapeutic and Diagnostic Agent for Glioblastoma Therapy and Detection: Alternative to Old Drugs and Agents

**DOI:** 10.3390/ph14111193

**Published:** 2021-11-20

**Authors:** Ahmet Kaynak, Harold W. Davis, Subrahmanya D. Vallabhapurapu, Koon Y. Pak, Brian D. Gray, Xiaoyang Qi

**Affiliations:** 1Division of Hematology/Oncology, Department of Internal Medicine, University of Cincinnati College of Medicine, and Brain Tumor Center at UC Neuroscience Institute, 3512 Eden Avenue, Cincinnati, OH 45267, USA; kaynakat@mail.uc.edu (A.K.); harold.davis19@gmail.com (H.W.D.); vallabsa@ucmail.uc.edu (S.D.V.); 2Department of Biomedical Engineering, College of Engineering and Applied Science, University of Cincinnati, Cincinnati, OH 45221, USA; 3Molecular Targeting Technologies, Inc., West Chester, PA 19380, USA; cpak@mtarget.com (K.Y.P.); Briangray@mtarget.com (B.D.G.)

**Keywords:** brain cancer, glioblastoma multiforme, saposin C, dioleoylphosphatidylserine, SapC–DOPS nanovesicle, phosphatidylserine-targeted therapy, chemotherapy, radiation, combinational treatment

## Abstract

Glioblastoma multiforme (GBM), the most common type of brain cancer, is extremely aggressive and has a dreadful prognosis. GBM comprises 60% of adult brain tumors and the 5 year survival rate of GBM patients is only 4.3%. Standard-of-care treatment includes maximal surgical removal of the tumor in combination with radiation and temozolomide (TMZ) chemotherapy. TMZ is the “gold-standard” chemotherapy for patients suffering from GBM. However, the median survival is only about 12 to 18 months with this protocol. Consequently, there is a critical need to develop new therapeutic options for treatment of GBM. Nanomaterials have unique properties as multifunctional platforms for brain tumor therapy and diagnosis. As one of the nanomaterials, lipid-based nanocarriers are capable of delivering chemotherapeutics and imaging agents to tumor sites by enhancing the permeability of the compound through the blood–brain barrier, which makes them ideal for GBM therapy and imaging. Nanocarriers also can be used for delivery of radiosensitizers to the tumor to enhance the efficacy of the radiation therapy. Previously, high-atomic-number element-containing particles such as gold nanoparticles and liposomes have been used as radiosensitizers. SapC–DOPS, a protein-based liposomal drug comprising the lipid, dioleoylphosphatidylserine (DOPS), and the protein, saposin C (SapC), has been shown to be effective for treatment of a variety of cancers in small animals, including GBM. SapC–DOPS also has the unique ability to be used as a carrier for delivery of radiotheranostic agents for nuclear imaging and radiotherapeutic purposes. These unique properties make tumor-targeting proteo-liposome nanocarriers novel therapeutic and diagnostic alternatives to traditional chemotherapeutics and imaging agents. This article reviews various treatment modalities including nanolipid-based delivery and therapeutic systems used in preclinical and clinical trial settings for GBM treatment and detection.

## 1. Introduction

Every year, around 25,000 new patients are diagnosed with primary malignant brain tumors, which have a poor prognosis with an overall 5 year survival rate of 36%, which drops to less than 22% for those over 40 [1]. On the other hand, nonmalignant brain and other central nervous system (CNS) tumors are much more common (~70% of brain cancers) and have a much brighter diagnosis of greater than 90% 5 year survival. Brain tumors are the eighth most common cancer overall among persons over 40 years and the third leading cause of cancer-related death in this population [2]. The most prevalent primary brain tumors in adults are nonmalignant meningiomas and malignant gliomas (such as glioblastoma, oligodendrogliomas, ependymomas, and astrocytomas) [2]. Of the latter, glioblastoma multiforme (GBM) is the most common (nearly 50% of malignant brain cancers) and the most aggressive [2,3,4]. While cancers originating elsewhere may metastasize to the brain and become secondary tumors, primary brain tumors behave differently, and, while they may migrate within the brain, they rarely spread outside of the CNS [2,3,4].

The grade of the tumor and the ability to resect it guide treatment decisions. Surgery depends on the tumor’s accessibility, size, and extent, as well as the patient’s overall health. The severity of primary brain tumors is differentiated into Grades I–IV (see Table 1) [5].

Secondary brain tumors, which have metastasized to the brain from a tumor in another location (usually lung, breast, colon, kidney, thyroid, or uterine cancers or melanoma) are much more common than primary brain tumors [3,6]. Ironically, these tumors are becoming increasingly more frequent as therapies for primary tumors improve to allow patients to live longer, giving the original cancer more time to spread to the brain. Therapeutic efficacy for these brain metastases is generally determined by the response of the primary cancer to treatment [3,6].

GBM is derived from astrocytes or oligodendrocytes that support nerve cells [5]. Glioblastoma can occur at any age but is more common in older adults, and it is usually diagnosed after patients complain of progressively worsening headaches, vision, nausea, vomiting, or seizures. The tumors are largely drug-resistant [7,8], and therapy is impeded by protection of the tumors behind the blood–brain barrier (BBB) [9,10,11]. The BBB protects the brain by restricting the passage of potentially harmful molecules but also potentially beneficial chemotherapeutic drugs. While small chemotherapeutic agents of less than 400 Da molecular weight with fewer than eight hydrogen bonds can passively cross the BBB [12], larger molecules and drugs fail to traverse the barrier, which makes it difficult to benefit from their therapeutic effects [13].

In comparison to old drugs and agents, nanocarriers show great promise for GBM therapy and diagnosis due to their distinctive properties such as size, shape, and surface properties. These characteristics can be easily adjusted to effectively carry and deliver therapeutic and imaging compounds directly and specifically to tumor cells and tissues in a controlled way with fewer side-effects. Nanosize carriers are highly preferable because they readily cross the BBB. They also increase the absorption, delay excretion, and decrease the uptake and removal of the chemotherapeutic or imaging agent from the circulation via the reticuloendothelial system. Furthermore, recently developed nanolipids such as saposin C (SapC) coupled with dioleoylphosphatidylserine (DOPS) have multiple advantages of carrying imaging agents and chemotherapeutic agent/radiation, as well as being drug by itself, in addition to its ability to cross the BBB for targeting GBM.

## 2. Standard Diagnosis and Treatment of GBM

The first approach when diagnosing a patient suspected of harboring GBM is magnetic resonance imaging (MRI), which can determine the size, shape, and location of the tumor. There are significant limitations in conventional MRI in differentiating benign from malignant tumors since this technique basically recognizes BBB defects, mass effects, and edema that can equivalently accompany neoplastic and non-neoplastic lesions [14]. Dynamic susceptibility contrast and diffusion-weighted MRIs are advanced techniques that may provide some further information such as discrimination between GBM and malignant lymphoma [15]. In order to diminish the drawbacks of conventional MRI, positron emission tomography (PET)–computational tomography (CT) is used as a further test for suspected GBM [16].

The FDA approved interstitial wafers, consisting of 3.8% carmustine, for the treatment of recurrent high-grade glioma in 1996 and primary glioma in 2004. Carmustine interstitial wafers implanted after resection of tumor increased the median survival by about 2 months and reduced the risk of death by about 30% over 30 months [17]. In the 2000s, interest shifted to temozolomide (TMZ), which can be administered orally rather than intravenously. TMZ causes double-strand DNA damage, thus leading to cell-cycle arrest between the G1 and S phases and cell death [18]. In 2005, Stupp et al. [19] showed the supremacy of TMZ combined with radiotherapy after surgery over surgery or radiation alone, and this became the standard treatment for GBM. Despite these advances, GBM prognosis is still bleak, with a median survival of only 1–1.5 years [20]. Although TMZ is transported through the BBB, its cytotoxic effects can be neutralized by various DNA repair mechanisms, reinforcing the structural stability of methylated DNA bases before the drug can cause extensive tumor cell death [12].

GBM cells are generally widely spread throughout the brain at diagnosis. While surgery is the first approach for GBM treatment, which can result in a 99% reduction of the primary tumor, TMZ and radiation are only marginally effective in preventing disease progression. Therefore, despite the total resection of obvious cancer in most patients, there is a recurrence of tumor either at the site where the tumor initiated or at more distal locations within the body.

Radiotherapy is a standard modality for a large group of cancers and is used either alone or in combination with chemotherapy, both before and after surgery [21,22,23]. Radiation acts by disrupting DNA synthesis, thus shrinking the tumor, inhibiting metastasis, or preventing the cancer from coming back. However, over the years, there has been an increase in data suggesting that, in some cases, low doses of radiotherapy may boost the growth and spread of some tumors [24], which results in more resistant cancer cells against subsequent doses of radiation [25,26,27,28]. Although most studies revealed a slight benefit from the addition of chemotherapy for GBM, a considerable number of clinical trials comparing standard radiation vs. radiation combined with TMZ have shown that TMZ provided a modest improvement with the median survival increasing from 12.1 to 14.6 months [19,21]. Therefore, this combination therapy is now standard of care for most cases of GBM.

## 3. Nanocarrier as a Drug Delivery System for GBM Therapy

Drugs used in GBM treatment are not entirely effective due to several factors. These include the BBB that prevents entrance of drugs into GBM tumors, macrophages that engulf drug molecules, and lack of specific targeting mechanisms for drug molecules to reach GBM tumor cells [29].

Over the last 10 years, many nanocarriers such as liposomes, micelles, different nanoparticles, or nanogels have been investigated as carriers for diagnosis and treatment of GBM. The use of lipid-based nanocarriers to deliver chemotherapeutic agents is an emerging concept, and, while nanovesicles have a natural tendency to cross the BBB, specificity is usually lacking [30,31]; thus, the drug may not be fully effective at the tumor site. Not only do these novel agents have to cross the BBB, but they must also contend with the tumor microenvironment (TME) which solid tumors generate to support their preservation by inhibiting key surveillance functions of the immune system. Moreover, the TME is very acidic due to tumors utilizing Warburg metabolism [32,33], which can suppress the response to chemo-, radio-, and immunotherapy agents [34,35]. Therefore, the GBM TME is one of the key factors for the progression of cancer and overall patient survival. Manipulation of the surface of the liposomes may enable them to deliver the anticancer agents specifically to the tumor site.

Tumor-targeted nanodelivery can be established through either passive or active means. Passive targeted delivery can be achieved through an enhanced permeability (EPR) effect. As the tumor grows, blood vessel formation to the tumor is stimulated, and tumor vasculature becomes leaky. This leads to an increase in the EPR effect. Additionally, this may lead to accumulation of macromolecules and nanocarriers near the tumor site [36]. The permeability of nanoparticles depends on physical parameters such as size, shape, surface charge, and hydrophobicity. Active tumor-targeted delivery can be achieved through the recognition of surface receptors on GBM tumor cells. Once targeting ligands on nanocarriers bind to GBM tumor cell receptors, nanocarriers can be internalized through receptor-mediated endocytosis. Several targets on GBM have been identified over the years. Some of the key surface receptors identified in GBM include epidermal growth factor receptor (EGFR), interleukin-13 receptor, αvβ3 integrin, glucose transporter(s), and transferrin receptors [17]. Nonetheless, many receptor-targeted nanodelivery systems have not yet improved efficacy, in terms of disease progression and survival.

The novel ligand-targeted delivery of drugs with liposomes not only enhances the active compound concentration near the tumor region but also reduces systemic toxicity. Kim et al. [37] encapsulated TMZ into cationic liposomes and modified the surface of liposomes with an anti-transferrin receptor mAb fragment to specifically target the GBM tumor. They showed that the effect of TMZ was boosted 10-fold with the liposome encapsulated form compared to free TMZ [37]. Huang et al. [38] designed a tripeptide sequence NGR-modified liposome which selectively binds to aminopeptidase N, a zinc metalloenzyme overexpressed on the surface of cancer cells. NGR-modified liposomes were loaded with comretastatin A4 (CA4). CA4 as a vascular disrupting agent inhibits angiogenesis by targeting the tumor vasculature. Its antitumor effects are due to prevention of Akt activation and microtubule polymerization. The inhibited Akt leads to a decrease in cell proliferation through arrest of the cell cycle. U87-MG orthotopic tumor-bearing mice treated with NGR-modified liposomes containing CA4 (NGR-SSL-CA4) had longer median survival (25 days) than mice treated with CA4-containing liposomes (SSL-CA4; 20.5 days) or CA4 alone (19 days). U87-MG cell migration and vasculogenic mimicry were evaluated with in vitro wound healing and matrigel-based tube formation assays, respectively. At 10 nM, NGR-SSL-CA4 showed similar inhibition of migration and tube formation to free CA4, while SSL-CA4 activity was lower, indicating that the NGR-SSL does not diminish the effects of CA4 [38]. In another study, Jiao et al. [39] established a liposome nanocarrier system with Pep-1 ligand, which specifically binds to the Interleukin-13 receptor α2, a plasma protein overexpressed in glioblastoma multiforme. Notably, cellular uptake was enhanced from 47.5% to 89.8% once Pep-1 was conjugated to the liposome surface. Functionalized liposomes loaded with the anticancer agent cilengitide (CGT), a peptide drug which inhibits integrin receptors αvβ3 and αvβ5, prevented cell adhesion and induced apoptosis of GBM cells, which express αvβ3 and αvβ5 receptors. CGT-loaded Pep-1 peptide-conjugated liposomes (PeCNL) showed a higher cytotoxic effect (IC_50_: 2.38 µg/mL) than CGT alone (IC_50_: 6.85 µg/mL) in U87 GBM cells. In a U87 xenograft mice model, PeCNL exhibited a strong reduction in tumor progression with a final tumor volume of ~350 mm^3^ compared to mice treated with CGT alone (~1250 mm^3^) or nontreated mice (~1600 mm^3^) [39].

Several therapeutic agents based on liposomes are currently in clinical trials. The ClinicalTrials.gov database was used to obtain liposome-based anticancer drug delivery systems for GBM (Table 2). Doxorubicin-encapsulated polyethylene glycolated (PEGylated) liposomes, in addition to TMZ, were used in a Phase II clinical trial for 63 patients with newly diagnosed GBM (NCT00944801). The results showed that the toxicity of the combination of PEG-Dox and TMZ was tolerable. However, addition of PEG-Dox did not enhance the efficacy of radiotherapy and TMZ [40]. In another clinical trial, nanoliposomal CPT-11 (liposomal irinotecan) completed Phase I/II studies with 34 high-grade glioma-suffering patients (NCT00734682) [41]. In another clinical trial setting, SapC–DOPS (BXQ-350) completed a Phase I study with 86 participants with patients suffering from advanced solid tumors and high-grade glioma (NCT02859857) and showed no dose-limiting toxicity (DLT) at the highest administered dose [42]. Another lipid-based drug, Ar-C (DepoCyt), began a Phase I/II study for 12 participants with recurrent GBM (NCT01044966). However, the study was terminated due to insufficient patient enrolment in the trial [43]. Rhenium-186 nanoliposomes (186RNL) were used in a Phase I/II study of 55 participants with recurrent GBM (NCT01906385). The Phase I study results indicated that the drug was well tolerated with no dose-limiting toxicity [44]. Liposomal vertoporfin (Visudyne), which is approved by the FDA for the treatment of an eye disease, was used in a Phase I/II study to test the dose escalation and efficacy in high-grade EGFR-mutated GBM. The trial is still in progress (NCT04590664) [45]. In another Phase I trial, mRNA-loaded liposomes (RNA-LP) were employed for newly diagnosed pediatric high-grade gliomas and adult GBM (NCT04573140) [46]. Recently, SapC–DOPS (BXQ-350) was used in a Phase I trial for newly diagnosed diffuse intrinsic pontine glioma or diffuse midline glioma (NCT04771897) in children. This trial is also still in progress [47].

## 4. Application of Nanodrugs as Radiosensitizers for GBM Therapy

Radiotherapy remains the most accepted treatment modality for a number of tumor types. Approximately, 50% of all cancer patients receive radiotherapy during their treatment. With improved technology and a better understanding of the mechanisms of radiation, radiotherapy is increasingly used as a standard therapy for many cancers, but there are still challenges to perfect radiotherapy as a curative modality. One of the main problems is that cancer cells become radioresistant, partially due to the hypoxia of the TME. Hypoxia-induced factor-1 is activated under hypoxic conditions, leading to activation of genes associated with tumor invasion and angiogenesis, and it promotes tumor resistance to radiotherapy [48]. Therefore, using radiosensitizers to make radiotherapy more effective at lower doses to ameliorate harmful effects on normal tissues has been proclaimed as a logical approach. One stratagem is to use oxygen-mimicking compounds which have been shown as potent radiosensitizers for several cancers. It has also been shown that some molecules which can carry oxygen such as hydrogen peroxide or nitric acid can be used as radiosensitizers [49,50].

Another tactic to enhance the efficacy of radiation is the use of high-atomic-number (Z) elements. Z elements improve radiation efficacy by absorbing high X-ray energy. Gold nanoparticles are considered potent tumor radiosensitizers due to their higher mass energy absorption capacity compared to soft tissues. Their properties, such as a high surface-area-to-volume ratio, being an inert material with high biocompatibility, having low toxicity, and low systemic clearance with a prolonged circulation time, make them strong candidates as radiosensitizers. Joh et al. tested the radiosensitization capacity of gold nanoparticles on U251 GBM cells. These cells, irradiated in the presence of 1 mM gold nanoparticles had a 1.7-fold higher γH2AX density, a marker for DNA damage, than those treated with radiotherapy alone. In an orthotopic GBM xenograft mouse model, gold nanoparticles followed by radiation prolonged the median survival to an average of 28 days compared to radiotherapy only (14 days, *p* = 0.011) [51]. Surface modification and functionalization are required to actively target the gold nanoparticles to the tumor. These modifications could enhance the absorption of nanoparticles, allowing for more precise control of biodistribution and higher accumulation at the tumor site.

Folate receptors are a particular interest for targeted delivery in GBM since one of the folate receptors, FRα, is highly expressed in gliomas but is very low on normal cells [52]. Kefayet et al. [53] prepared folic acid- and BSA-coupled gold nanoclusters (FA-AuNCs) to explore their radiosensitizing capacity on C6 glioma tumors. They used ICP-OES to evaluate the targeting efficacy of FA-AuNCs and demonstrated that FA-AuNCs had a 2.5-fold higher accumulation in cancer cells than in normal cells. Moreover, in a C6 glioma intracranial tumor model, FA-AuNC concentration was higher in the tumor region compared to normal brain tissue, and a combination of FA-AuNCs with radiotherapy (25.0 ± 1.5 days) increased survival time by 7 days (*p* < 0.001) compared to radiation therapy alone (18.3 ± 1 day) [53].

Monoclonal antibody (mAb)-conjugated gold nanoparticles have also been used to enhance targeted delivery of radiosensitizers. Groysbeck et al. [54] synthesized cetuximab, an anti-epidermal growth factor receptor (EGFR) mAb, and then conjugated it with gold nanoparticles to develop a targeted glioblastoma radiosensitizer (Au-Cmab). Immunofluorescence staining showed that Au-Cmab accumulates on EGFR (+) U87 cells but not on negative control, EGFR (−) U87 cells.

## 5. Radiolabeled Nanodrugs for GBM Therapy and Imaging

Radioisotopes emit ionized atoms and free radicals by releasing high energy from the nucleus, which leads to DNA damage in cells. Beta emitters such as ^188^Re, ^186^Re, ^89^Sr, ^32^P, and ^90^Y as well as alpha-emitters such as ^213^Bi, ^211^At, and ^225^Ac, are commonly used radioisotopes for radiation oncology. Radioisotopes are quickly eliminated from the body via the kidneys or by the reticuloendothelial system after engulfment by phagocytes. However, by loading them into nanoparticles, radioisotopes can escape these clearance systems. Moreover, radioisotope-labeled nanoparticles can be used to increase accumulation at the tumor site and decrease undesirable biodistribution [55]. Phillips et al. [56] developed rhenium-186-encapsulated liposomes and evaluated their efficacy by convection-enhanced delivery in an orthotopic U87 glioma rat model. They showed that doses up to 1840 Gy were achieved without toxicity. The median survival of ^186^Re-liposome-treated animals was significantly higher compared to controls (mean survival of 126 vs. 49 days; *p* = 0.0013) [56].

Some modifications of nanoparticles, such as PEGylation generate steric hindrance, which prevents renal clearance of the particles to increase the half-life of radioisotopes in the blood. Huang et al. [57] conjugated ^188^Re with PEGylated nanoliposomes and evaluated the therapeutic effect on GBM in orthotopic glioma-bearing mice. This radioisotope-labeled nanoliposome prolonged the survival of the mice by 10.67% compared to the saline-injected control group (mean survival of 20 vs. 18 days; *p* < 0.05) [57].

Combination of radiotherapy with chemotherapy is one of the most effective treatment modalities for GBM. Nanotechnology can improve chemoradiotherapy in two ways. One is to carry the chemotherapeutics by nanodrugs which have radiosensitizing effects when combined with external irradiation [58]. A number of identified chemotherapeutic agents have been used as radiosensitizers, and they have favorably improved the efficacy of radiotherapy for various cancers in clinical trials [59]. For instance, gemcitabine is also an efficient radiosensitizer in the treatment of many cancers, such as thyroid cancer, pancreatic cancer, and sarcoma [60,61,62]. While many chemotherapeutic agents have been investigated as potential radiosensitizers for GBM, most have failed in clinical trials.

Another option is to co-deliver the chemotherapeutic agent with radioisotopes in the same particle. Gao et al. [63] used ^131^I-labeled doxorubicin-loaded nanoliposomes (^131^I-DOX-NL) to evaluate the cytotoxic effect on tumors in a U87 xenograft model. Doxorubicin is a chemotherapeutic drug which inhibits topoisomerase II, an enzyme which relaxes supercoiled DNA for transcription. Mice treated with ^131^I-DOX-NL had about 50% longer mean survival time compared to those treated with Na^131^I (46 days vs. 31 days, *p* = 0.0005) or DOX alone (30 days, *p* = 0.0004) and about 20% compared to those treated with liposome-loaded ^131^I (38 days, *p* = 0.0308) or liposome-loaded DOX (36 days, *p* = 0.015). These results indicate that combined radiotherapy and chemotherapy has enhanced antitumor activity and can significantly improve survival [63].

In addition to being a drug carrier, nanodrugs can also be used for bioimaging by labeling with radionuclides or contrast agents. Oku et al. [64] reported the feasibility of positron emission tomography (PET) imaging using both free, radiolabeled 1-[^18^F]fluoro-3,6-dioxatetracosane ([^18^F] FDG) and liposome-incorporated [^18^F] FDG. ^18^F-labeled liposomes were superior to free [^18^F] FDG in a rat glioma model, capable of detecting very small tumors (1 mm) with quite low background signal. Biodistribution studies showed that injected free [^18^F] FDG was primarily taken up by the heart and normal regions of brain and degraded quickly. However, liposomal [^18^F] FDG was absorbed by the glioma and then maintained in the bloodstream before accumulation in the spleen prior to excretion. Malinge et al. [65] used ^68^Ga-labeled magnetic liposomes as a dual imaging modality and successfully visualized U87-MG tumors in Swiss nude mice using PET and magnetic resonance imaging (MRI). Huang et al. [66] applied a beta emitter, ^188^Re-labeled PEGylated liposomes, to image GBM. The maximum concentration of ^188^Re-liposome with 1.95% injected dose (ID)/g was reached in the tumor 24 h after intravenous injection, whereas the accumulation was much lower in the normal brain (0.06% ID/g at 24 h post injection; ratio of tumor to normal brain uptake = 32.5). A clear tumor image was taken from 4 h until 48 h with SPECT/CT.

Below, we discuss a tumor-targeted nanovesicle called SapC–DOPS that exhibits robust tumor targeting and intrinsic cancer cytotoxicity.

## 6. SapC–DOPS Nanovesicles for Precise Targeting of Brain Tumors

Phospholipids are located asymmetrically in cell membranes. While neutral phospholipids are found on the outer leaflet, anionic lipids such as phosphatidylethanolamine and phosphatidylserine (PS) are located on the inner side of the membrane [67,68,69]. Many cancer cells, unlike normal cells, expose high amounts of PS on the outer leaflet of their plasma membrane. Taking advantage of this characteristic, an anticancer drug was developed which comprises the fusogenic protein, SapC and DOPS vesicles. SapC–DOPS nanovesicles specifically target several cancer cells and lead to apoptotic cell death.

SapC is a ubiquitous lysosomal protein, which has high affinity and striking specificity for PS. It functions to catabolize glycosphingolipids [70]. Once SapC is combined with DOPS, approximately 200 nm stable nanovesicles are formed which selectively fuse with the surface PS on cancer cells. SapC–DOPS selectively targets PS on the tumor cell surface, and, unlike most other therapeutics, its activity is enhanced by the acidic TME. SapC–DOPS binding to the cancer cells leads to ceramide accumulation, caspase activation, and eventual apoptosis [71] via a variety of mechanisms (Figure 1). Importantly, astrocytes in coculture with GBM or metastatic breast cancer cells are not targeted or killed by SapC–DOPS and, critically, do not protect the tumor cells from the effects of SapC–DOPS [72].

Many solid brain tumors have abnormal angiogenesis and increased vascular permeability. This may allow SapC–DOPS to access the tumor. Blanco et al. used fluorescently labeled markers to reveal an enlarged, irregular tumor vasculature with increased permeability. The results demonstrated that SapC–DOPS nanovesicles are capable of crossing the blood–brain–tumor barrier (BBTB) [72]. Regardless of the conduit, SapC–DOPS selectively and effectively crosses into the brain parenchyma to target and prevent growth of primary tumors (neuroblastoma xenografts and orthotopic glioblastomas) and secondary brain metastases derived from human breast or lung cancer cells, in mice [72]. This is contingent on cell surface PS, as lactadherin, which binds PS with high affinity, blocked SapC–DOPS binding and tumor cell death [72]. In addition to the tumor toxicity, SapC–DOPS exerts a strong antiangiogenic activity, and, unlike traditional chemotherapies, hypoxic cells are sensitized to SapC–DOPS-mediated killing [73].

To determine efficacy, survival of mice with GBM in two different tumor models was compared after SapC–DOPS treatment. U87ΔEGFR cells grow quickly, and the mean survival was ~13 days for untreated mice and 18 days for SapC–DOPS-treated mice (*p* < 0.0001). All of the untreated mice died by 18 days, while 25% of the treated mice lived until they were euthanized at 350 days. For a long-term model, X12v2 cells were used. In this case the mean survivals were 80 and 128 days for untreated and treated, respectively (*p* < 0.0001). All of the untreated mice died by day 128, while 75% of the treated mice lived until euthanized at 250 days. Likewise, SapC–DOPS selectively targets brain metastasis-forming cancer cells in mouse models and prolongs the survival of mice harboring brain metastases [72]. As discussed more comprehensively below, Wojton et al. [73] showed that SapC–DOPS is more effective against cells with high surface PS; for example, X12v2 tumors with high surface PS are more susceptible to SapC–DOPS than U87ΔEGFR tumors with relatively low surface PS [73].

GBM is known for its inherent and acquired resistance to chemotherapies, including to TMZ and to radiotherapy [74,75]. However, Wojton et al. [76] examined the effects of a treatment regime of suboptimal SapC–DOPS alone or in combination with TMZ in vitro and in mice bearing GBM, and they evaluated the effect of SapC–DOPS in vitro in combination with TMZ using Chou–Talalay analysis. The results demonstrated that SapC–DOPS shows synergism with TMZ (Table 3) [76]. Tumor presence and size was evaluated in mice selected for various treatments by T2-weighted magnetic resonance imaging (MRI), and visible tumors were detected in all of the control mice, 60% of the TMZ-treated mice, 80% of the SapC–DOPS-treated mice, and none of the combination-treated mice at day 48 post tumor implantation; furthermore, there was a significant enhancement in median survival of mice compared to either agent alone [76]. SapC–DOPS kills tumors cells without apparent off-target toxicity to normal cells and tissues [72,73].

A bath sonication method was used for laboratory-scale preparation of SapC–DOPS. In larger-scale manufacturing, a lyophilization procedure with an organic solvent-water system was used for SapC–DOPS nanodrug preparation. In mice, no significant difference was observed in terms of pharmacokinetics profile of SapC–DOPS prepared with either preparation method. The size of SapC–DOPS nanovesicles prepared using both methods was also comparable. In fact, larger-scale preparations of SapC–DOPS nanovesicles (BXQ-350) were successfully prepared and tested in human clinical trials. BXQ-350 has successfully completed a Phase 1 clinical trial and was well tolerated without any dose-limiting toxicity [42].

## 7. SapC–DOPS Nanovesicles for Tumor Detection

SapC–DOPS labeled with a far-red fluorescent probe, radioactive tags, or contrast agents has been used to distinguish neoplastic tumor regions for detection of mouse brain tumors in vivo by nuclear imaging, magnetic resonance imaging, and multiangle rotational optical imaging [77,78]. SapC–DOPS–CVM injected into mice was detected within 30 min and remained within the tumor for at least 7 days, whereas nontumor tissues were unstained. SapC–DOPS has also been used as a carrier for MRI contrast agents. Ferumoxtran-10, an ultrasmall superparamagnetic iron oxide (USPIO) contrast agent, was encapsulated into SapC–DOPS vesicles to detect tumors in mice using MRI [77]. The T2 relaxation time (i.e., time for the transverse magnetization to fall to 37% of its initial value after magnetization) of subcutaneous neuroblastoma xenografts was decreased by SapC–DOPS–USPIO indicating its uptake and allowing detection of the tumor. The paramagnetic contrast agent, gadolinium, was also incorporated into SapC–DOPS vesicles and produced a 9% increase in the longitudinal relaxation rate (R1) of orthotopic GBM tumors in mice within 10 h of injection, with only minimal changes in normal brain tissue [79]. In addition, CMV derivatives have been used to label SapC–DOPS with ^124^I, a positron emitter, and these were used to selectively enhance the intracranial glioblastomas with PET scanning [78].

As mentioned, radiation is a first-line brain tumor treatment. Recent studies showed that radiation induces a rapid and consistent increase in surface PS on a variety of cancer cells with initially low to moderate surface PS [28]. Although this could be partially due to early apoptosis, these cells continue to proliferate with higher surface PS [28]. However, it has been shown that cancer cell lines are heterogeneous for surface PS [73], and it is likely that radiation preferentially kills the subpopulation of cells with lower surface PS (Figure 2), thus augmenting the mean surface PS of the remaining population. This increase in surface PS facilitates the extermination of the cells by SapC–DOPS in vitro and in subcutaneous tumors [28]. Interestingly, radiation also increases surface PS on tumor blood vessels, permitting employment of the PS-targeting antibody, bavituximab, for successful treatment of lung cancer and glioblastomas in mice [80,81,82]. Crucially, it has been demonstrated that radiation did not increase surface PS exposure on normal cells [28], and no significant cytotoxic effect of SapC–DOPS on normal cells was observed.

## 8. SapC–DOPS as a Carrier for Radioisotopes

SapC–DOPS nanovesicles meet the qualifications for carrying fluorescent probes and MRI contrast agents selectively into the tumor tissue when administered intravenously [77,78,79]. Hence, Blanco et al. [78] tested whether SapC–DOPS could serve as a delivery agent for radiation and, thus, contribute as a cytotoxic drug and radiotherapy in one modality. Although free iodine is targeted only to the thyroid, radioactive iodine isotopes kill all cells. Therefore, limiting the exposure of radioactive iodine to normal cells is important. Blanco et al. [78] demonstrated that ^125^I-coupled SapC–DOPS not only blocks thyroid uptake but also rapidly (within 24 h) leads to radioisotope accumulation in brain tumors.

Next, a carbocyanine-based fluorescent probe [DiD(16,16)] with ^131^I was incorporated into SapC–DOPS nanovesicles (^131^I-DiD(16,16)–SapC–DOPS, Figure 1) [71]. ^131^I is of great interest for both radiotherapy and imaging applications and was selected for this study because (1) it is an FDA-approved radionuclide for treatment of thyroid cancer [82], (2) it has a short half-life (eight days), and (3) it emits both β and γ radiation. Treatment with β-particle-emitting radionuclides is the favored way to control large and nonhomogeneous tumors [83,84]. ^131^I-labeled SapC–DOPS nanovesicles may be an effective weapon for personalized and targeted brain cancer therapy by taking the advantages of SapC–DOPS which successfully passes through the BBTB and selectively targets PS on the tumor surface. ^131^I-DiD(16,16)–SapC–DOPS exhibited a time to peak in the blood of 4 h and a bioelimination half-life of 11.5 h. To determine the efficacy of the compound, GBM cells were implanted into the right frontal cortex just lateral of the sagittal suture of nude mouse brains. The mice were treated with PBS or ^127^I-DiD(16,16)–SapC–DOPS (nonradioactive isotope of iodine) or ^131^I-DiD(16,16)–SapC–DOPS. In radiation studies, a dose of SapC–DOPS with only a modest benefit was picked to determine whether targeted dosing of ^131^I would increase survival could be determined. Indeed, ^131^I-DiD(16,16)–SapC–DOPS (median survival ID_50_ = 20 days)-treated mice had longer survival than the mice treated with ^127^I-DiD(16,16)–SapC–DOPS (ID_50_ = 14 days; improvement of >43%; *p* = 0.0378) or PBS (ID_50_ = 13 days; improvement of >48%, *p* = 0.0004) [71]. Furthermore, the surface PS increase induced by radiation may further enhance the tumor cytotoxicity of SapC–DOPS. Future studies should focus on finding effective ^131^I dosage to be used in combination therapy with minimal radiation-induced toxicity. Indeed, once initial radiation effects are built up, ^131^I can be removed from the dosing regimen since it has been shown by a number of researchers that recurrent radiation treatment induces radiation resistance [26,28].

Tumors commonly display heterogeneous characteristics such as excessive surface PS exposure [73,85]. Most cancers have elevated surface PS exposure, but this shows variability even within the same tumor cell line [69]. It has been demonstrated that SapC–DOPS specifically targets cells with higher surface PS which are predominantly in the G2 cell phase, whereas radiation has a better effect on cells with lower surface PS cells are mainly in the G1 cell phase [28]. Similarly, TMZ arrests cells in the more susceptible G2/M cell-cycle phase and enhances the DNA-damaging effects of radiation [86]. Therefore, combining these two modalities should lead to enhanced tumor cell death (Figure 2).

## 9. Conclusions

GBM is the most common primary brain tumor in adults with the lowest survival rate of all brain cancers despite multimodal treatment strategies using surgery and chemoradiotherapy [87]. Radiation therapy is an essential treatment modality for GBM, and technological advances in the field of radiation oncology have contributed to an enhanced radiation effect. Nanosize carriers are promising candidates for the delivery of radioisotopes with or without chemotherapeutic agents to the tumor site. Tumor-targeted delivery can be achieved by coupling nanocarriers with cancer targeting ligands. These platforms can also be used for the delivery of MRI or PET contrast agent to the tumor site. Recent studies demonstrated that proteo-liposome nanocarriers such as SapC–DOPS are capable of both tumor targeting and nuclear imaging with a good synergistic radio/chemo combination effect. In spite of all of the advances cited here, further efforts in preclinical and clinical settings are required to validate the use of radiolabeled nanocarriers for diagnostic and therapeutic purposes.

## Figures and Tables

**Figure 1 pharmaceuticals-14-01193-f001:**
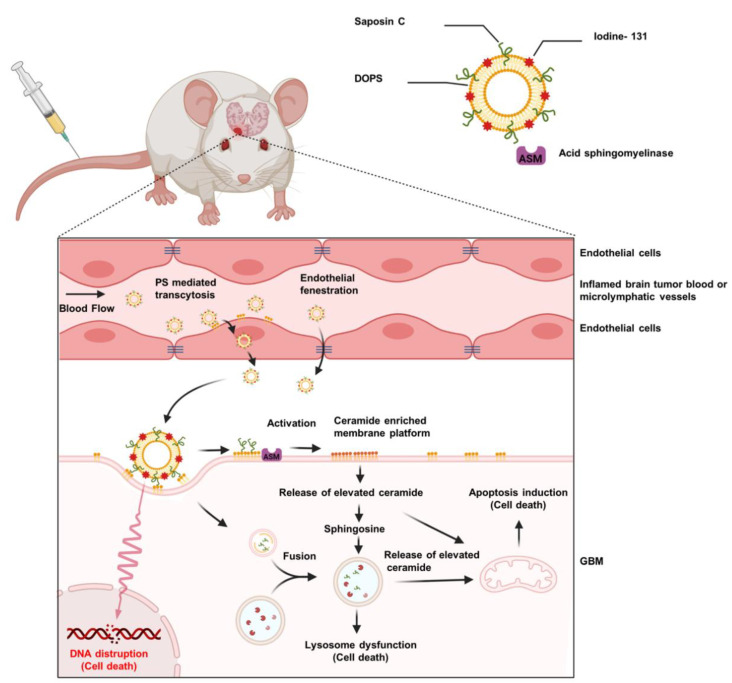
**Mechanism of action of SapC–DOPS and ^131^I-DiD(16,16)–SapC–DOPS.** SapC–DOPS selectively binds to the PS-rich membrane surface and is internalized into the cell through the endocytosis. After fusion with lysosomes, SapC leads to activation of acid sphingomyelinase and induces ceramide accumulation in the lysosome. Increasing ceramide induces cell death by causing lysosome membrane leakage and loss of mitochondrial membrane potential. Beta particles and gamma rays released from ^131^I cause DNA damage and induce cell death.

**Figure 2 pharmaceuticals-14-01193-f002:**
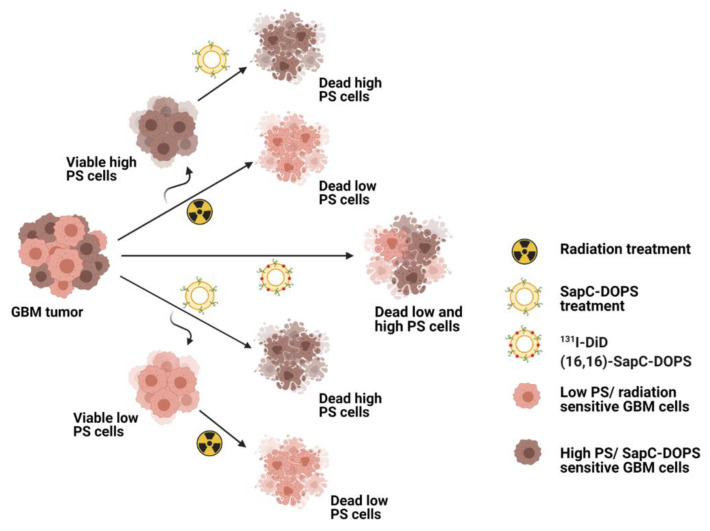
**SapC–DOPS and radiation kills the tumor cells in a PS-dependent manner.** PS exposure is heterogeneous within the tumor. Tumors consist of high-PS and low-PS cancer cells. Radiation (either radiotherapy or ^131^I-DiD(16,16)–SapC–DOPS), kills low-PS cancer cells and the leftover high-PS cancer cells are more sensitive to SapC–DOPS treatment. Conversely, SapC–DOPS preferentially target the high-PS cancer cells and leftover low-PS cancer cells are more sensitive to radiation. Thus, with combined or sequential treatment with SapC–DOPS and radiation, more cancer cells are eliminated.

**Table 1 pharmaceuticals-14-01193-t001:** Staging of glioblastoma.

Grade	Tumor
I	Tumor grows slowly and rarely spreads into adjacent brain or CNS tissue. Surgery is normally successful.
II	Tumor grows slowly but may spread into adjacent brain or CNS tissues and may recur following resection.
III	Tumor grows quickly, generally spreads, and its cells have an abnormal morphology.
IV	Tumor grows and spreads very quickly, and its cells have an abnormal morphology.

**Table 2 pharmaceuticals-14-01193-t002:** Therapeutic liposome nanovesicles studied for glioblastoma in clinical trials.

Clinical Trial Identifier	Trial Phase	Therapeutic Agent	Patient Profile	Number of Patient	Completion Status	Result	Ref.
NCT00944801	Phase I/II	PEGylated liposomal doxorubicin (PEG-Dox)	Newly diagnosed glioblastoma	63	Completed	Well-tolerated toxicity. No significant survival benefit from PEG-Dox combination with RT compared to TMZ combination with RT.	[40]
NCT00734682	Phase I	Nanoliposomal CPT-11 (liposomal irinotecan)	Recurrent high-grade gliomas	34	Completed	Not available.	[41]
NCT02859857	Phase I	SapC–DOPS (BXQ-350)	Solid tumors and glioma	86	Completed	The therapy was well tolerated; no dose-limiting toxicity or serious adverse events were observed.	[42]
NCT01044966	Phase I/II	Intraventricular liposomal encapsulated Ara-C (DepoCyt)	Recurrent glioblastoma	12	Terminated	The study was terminated due to lack of adequate patient enrolment into trial.	[43]
NCT01906385	Phase I/II	Rhenium-186 nanoliposome (186RNL)	Recurrent glioma	55	Recruiting	The therapy was well tolerated; no dose-limiting toxicity or serious adverse events were observed.	[44]
NCT04590664	Phase I/II	Liposomal verteporfin (Visudyne)	High-grade EGFR-mutated glioblastoma	24	Recruiting	Not available.	[45]
NCT04573140	Phase I	mRNA lipid particle (RNA-LP)	Newly diagnosed pediatric high-grade gliomas and adult glioblastoma	28	Recruiting	Not available.	[46]
NCT04771897	Phase I	SapC–DOPS (BXQ-350)	Newly diagnosed diffuse intrinsic pontine glioma or diffuse midline glioma	22	Recruiting	Not available.	[47]

**Table 3 pharmaceuticals-14-01193-t003:** Synergistic effect of the combination treatment with SapC–DOPS and TMZ for GBM cells.

	GBM Cells		
Fraction Affected (Fa)	X12v2	GBM169	Gli36EGFR
0.2	Strong synergy *	Synergy	Moderate synergy
0.4	Strong synergy	Strong synergy	Strong synergy
0.6	Strong synergy	Strong synergy	Strong synergy
0.8	Strong synergy	Synergy	Strong synergy

* Synergistic effect based on combination indices (CIs). Strong synergy CIs < 0.4; synergy CIs = 0.4–0.6; moderate synergy CIs = 0.6–0.8.

## Data Availability

Data sharing not applicable.

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
