# Peer review of "SapC–DOPS as a Novel Therapeutic and Diagnostic Agent for Glioblastoma Therapy and Detection: Alternative to Old Drugs and Agents"

_pharmaceuticals, 2021, doi:10.3390/ph14111193_

Round 1

Reviewer 1 Report

The work described in the present manuscript is consistent with the scope of the journal. SapC-DOPS as a Novel Therapeutic and Diagnostic Agent for Glioblastoma Therapy and Detection: Alternative to Old Drugs and Agents. This review work is methodically carried out and scientifically correct. The authors have discussed the novel in vivo and clinical approaches of radio-nanomedicine for GBM treatment and detection. Overall it’s a very good article and I highly recommend it. There are minor issues that the authors can address to improve their manuscript before acceptance for publication.

The novelty of the manuscript must be highlighted in the introduction section

The authors can include a table related to a clinical trials for Therapeutic and Diagnostic Agent studied for Glioblastoma Therapy

All figures have low resolutions. Authors should use high-resolution images.

Reviewer 2 Report

The authors have written a review about GBM and GBM treatments in general. The authors have given a very general review of previous GBM treatments finishing with a very specific review of their own works regarding SapC-DOPS. While the review is well written and clear major revisions are needed. First, the abstract only disucsses the authors own works, yet the first 7 pages are a review in general of GBM treatmens used in the past. Secondly the review is written in a first person way "We did....... our group examined" A review should look at the literature overall and not just self-cite. Therefore, the review should be rewritten, both to include the background etc, in the first 7 pages and to remove all direct self-references. Even if no other group works on these systems, direct self promotion should not be permitted. Finally, the conclusions are very specific only talking about a small portion of the results discussed and no expert opinion or future needs/missing pieces of these systems is discussed.

Round 2

Reviewer 2 Report

I thank the authors for making the requested adjustments